# The Quality Evaluation of Avocado Fruits (*Persea americana* Mill.) of Hass Produced in Different Localities on the Island of Tenerife, Spain

**DOI:** 10.3390/foods13071058

**Published:** 2024-03-29

**Authors:** Clemente Méndez Hernández, Alicja Grycz, Domingo Rios Mesa, Beatriz Rodríguez Galdón, Elena M. Rodríguez-Rodríguez

**Affiliations:** 1Servicio de Agricultura del Cabildo Insular de Tenerife, 38071 Tenerife, Spain; clementem@tenerife.es (C.M.H.); domingor@tenerife.es (D.R.M.); 2Departamento de Ingeniería Química y Tecnología Farmacéutica, Universidad de La Laguna, 38296 Tenerife, Spain; alicja.grycz@op.pl (A.G.); bgaldon@ull.edu.es (B.R.G.)

**Keywords:** Hass avocado, production area, antioxidant, mineral, fatty acid

## Abstract

The effect of the production area on the quality of Hass avocados grown on the island of Tenerife was studied. For this purpose, several physicochemical parameters, such as fruit weight, percentage of pulp, seed and skin, proximate composition, minerals, total phenolic compounds (TP), total flavonoid compounds (TF), α-tocopherol, antioxidant capacity, and fatty acid profile were analyzed. The location of the orchards significantly influenced avocado weight; pulp and seed percentage; and fat, fiber, ash, α-tocopherol, TP, phosphorus, potassium, calcium, iron, zinc, and oleic and palmitoleic acid contents. Buenavista (BU) avocados were the smallest (185 g) and presented the highest percentage of pulp (77.1%) and lowest percentage of fiber (5.43%). The highest levels of dry matter (33.8%) and fat (20.4%) were found in avocados harvested in Los Silos (SI) and Santiago del Teide (SA), respectively. Compared with those at the other locations, the avocados harvested in Güímar (GU) had high levels of α-tocopherol (52.2 µg g^−1^) and phenolic compounds (56.0 mg GAE 100 g^−1^). Avocados from Los Realejos (RE) had the highest percentage of oleic acid and the lowest percentage of palmitoleic acid. Numerous significant correlations were found between the variables studied, especially those between TP, TF, and antioxidant capacity (DPPH) and between fat percentage and dry matter.

## 1. Introduction

The avocado belongs to the *Lauraceae* family, and its fruit is a large berry with a large seed inside. The avocado is a plant well adapted to tropical and subtropical climates. Unlike other nonclimacteric fruits, avocado fruits can be harvested before they reach commercial maturity, as they can ripen after being harvested. It is cultivated mainly in Mexico, the United States, Australia, South Africa, and Spain [1,2]. There are more than 100 avocado varieties registered, although the most popular commercial varieties are Hass and Fuerte. The thick peel of the Hass variety makes it less susceptible to physical damage after harvest and during handling [3]. Avocado production and consumption have been increasing worldwide in the last 10 years [4]. For example, in the world market in 2012, the amount of avocado produced was close to 4.2 million tons (MT), and in 2022, it was close to 8.4 MT, representing a growth of 200% [4]. Avocado is a fruit rich in nutrients, especially oil (15–20% in fresh weight), and the predominant fatty acids are unsaturated, such as oleic acid (50–60% of total fat) and linoleic acid (11–15%) [5,6]. The carbohydrate content is low (8–10% of the pulp), and avocado is distinguished by its high content of heptoses-D-mannoheptulose (D-manno-2-ketoheptose), which plays a role as an antioxidant, and perseitol (D-glycero-D-galactoheptitol), a storage compound involved in the transport of micronutrients. In addition, it contains other sugars, such as sucrose, glucose, and fructose, in minor amounts [7,8]. Avocado fruits contain numerous bioactive compounds, such as vitamin E, phenols, flavonoids, carotenoids, and sterols, among others, which show antioxidant activity, preventing oxidative damage to the body’s cells by free radicals. In recent years, more attention has been given to the antioxidant content of food products, as several epidemiological studies have shown that antioxidants are associated with a reduction in the incidence of cardiovascular diseases, neurodegenerative diseases, and cancers [3,9,10,11]. Thus, phytochemicals present in avocados have been found to inhibit the growth of cancerous tumors, reduce obesity and diabetes, and mitigate cardiovascular and neurodegenerative diseases [3,9,10,11]. The consumption of certain avocado products, such as avocado oil, improves hepatic lipid metabolism and reduces the risk of atherosclerosis. Specifically, the consumption of palmitoleic acid could reduce elevated triglyceride and low-density lipoprotein (LDL) levels [11].

As a plant that requires tropical or subtropical climatic conditions, avocado grows well in the Canary Islands, with the Hass cultivar being the main one in the local market. The rate of avocado production in Tenerife is increasing; between 2007 and 2021, the cultivation area tripled, reaching 956 ha and representing 42% of the total cultivation area of this plant in the Canary Islands [12]. To date, the Canary Islands avocado is mainly present in the local market, but it is highly appreciated abroad for its flavor. The purpose of this study is a first step to establish the composition of avocados produced in Tenerife, in order to promote the consumption and export of the Canary Island Hass avocado in Europe and to create a quality brand to differentiate the Canary Island Hass avocado from those produced in competing countries. The island of Tenerife has numerous microclimates where avocados are grown, both on the northern and southern slopes and from sea level to almost 1000 m altitude. This means that avocados produced in different parts of the island have different characteristics and qualities depending on the growing areas. There are no clearly established criteria for the quality of Hass avocado produced in Tenerife. Therefore, in this study, the physicochemical characteristics and composition of Hass avocados from orchards located in different agroclimatic conditions of Tenerife were determined. The characteristics of the island of Tenerife, which has different cultivation heights and slopes, resemble those of a small continent, and its location in the subtropics means that the results can be extrapolated to the production of avocados at other latitudes in the face of climate change [13,14]. The evaluation of the quality of Tenerife avocados can be extrapolated to other islands producing this fruit in the Canary archipelago, such as La Palma, La Gomera, Gran Canaria, and El Hierro. Likewise, this study can complement those carried out in other subtropical locations with young volcanic soils and carbonate alkaline waters from the recharge of mountain aquifers.

## 2. Materials and Methods

### 2.1. Plant Material

A total of 50 avocado (*Persea americana* Mill.) fruit samples of the Hass variety were collected between January and March 2020 from different orchards located on the island of Tenerife, Spain. Samples were collected from 7 localities representative of Hass avocados on the island of Tenerife: Buenavista (BU) (n = 2), Los Silos (SI) (n = 2), La Orotava (OR) (n = 27), Güímar (GU) (n = 6), Los Realejos (RE) (n = 8), Santiago del Teide (SA) (n = 3), and Tacoronte (TA) (n = 2) (Figure 1). In each orchard, three homogeneous fruits were taken from three trees of similar maturity and without physical damage.

Table 1 shows the average monthly relative temperature and humidity, precipitation, and reference evapotranspiration data of 7 climatic stations representative of the areas where the orchards were located. Sampling was performed between 1 April 2019 and 31 March 2020 (the period that influenced the harvest from flowering to the end of collection). The slope of the island where it is located and the latitude, altitude, and climatic classification according to Papadakis [15] were determined in the agroclimatic study of the island carried out by Santana [16]. The samples collected in the different locations are also presented in Table 1. The irrigation water for each location of avocado orchards is shown in Appendix A.

### 2.2. Sample Preparation

The fruits were stored in ripening chambers at 20 °C and 95% RH (relative humidity) and were considered mature when the skin color reached color 6, as indicated by White et al. [17]. Once ripe, the Hass avocados were rinsed with distilled water and dried with a paper towel. From each sample, all avocados were weighed to calculate the average weight per unit. The skin and seeds of four avocados from each sample were removed, and the avocados were weighed separately to calculate the percentage of each part of the fruit.

For analyses that were performed immediately, such as ascorbic acid, total phenolic compounds, total flavonoids, antioxidant activity, moisture, and α-tocopherol and fatty acid profile, a 1 cm slice was cut from each fruit, the skin was removed, and the pulp was homogenized with a mortar. Several portions of this pulp were taken for analysis. Half of the rest of each fruit (without skin and seed) was also homogenized with a mortar, transferred to a laboratory glass dish, and then placed in a microwave oven (Proclen 6010 800 W, Cecotec Innovaciones s.l., Valencia, Spain) for drying to a final moisture content between 4% and 5%. The dried samples were stored in bags in a freezer at −20 °C. It was necessary to obtain approximately 20–30 g of each dried sample to perform the remaining analyses, such as total fat, dietary fiber, and minerals and trace elements. All analyses were performed in triplicate.

### 2.3. Proximate Composition

Moisture was determined by gravimetry by first drying the sample in a microwave oven (Proclean 6010 800 W, Cecotec Innovaciones s.l., Valencia, Spain) and then introducing it into an oven (Conterm 36 L, J.P. Selecta s.a.u., Barcelona, Spain) at 70 °C until a constant weight was obtained, and the dry matter content was calculated by difference. Ash was determined from the residue of moisture analysis by calcination in a muffle furnace (J.P. Selecta, S.a.u., Barcelona, Spain) until white ash was obtained, and the temperature was gradually increased to 550 °C [18]. Total fat was analyzed by the method described by Meyer and Terry [19]. Each sample (2 g dry sample) was subjected to fat extraction using n-hexane in a Soxhlet extraction system (Sox 414, C. Gerhardt GmbH and Co. KG, Konigswinter, Germany). Dietary fiber content was determined by an enzymatic-gravimetric assay (Megazyme International Ireland, Bray, Ireland) using a Fibertec 1023 analytical system (FOSS, Hillerod, Denmark) [19].

### 2.4. Ascorbic Acid Content

Ascorbic acid (Sigma-Aldrich, St. Louis, MO, USA) was analyzed by the indophenol 2,6-dichlorophenol (DIP) titration procedure [18]. Four grams of the avocado homogenate was mixed with 10 mL of metaphosphoric acid solution (3%) in an ultrasonic bath (Ultrasons 4 L, J.P. Selecta s.a.u., Barcelona, Spain) for 5 min and then filtered and titrated with DIP until the appearance of a stable pink color.

### 2.5. Mineral Content

The analysis of minerals and trace elements (potassium, K; calcium, Ca; magnesium, Mg; iron, Fe; copper, Cu; zinc, Zn; and manganese, Mn) was performed with a Varian SpectrAA 50B atomic absorption spectrometer (Varian Ibérica SL, Madrid, Spain), and the analysis of phosphorus (P) was performed via the colorimetric method with vanadate-molybdate reagent (Panreac Química s.a., Barcelona, Spain). The samples (2 g of dry sample) were previously digested with 10 mL of nitric acid Hyperpur (Panreac Química s.a., Barcelona, Spain) according to the procedure described by Hernández Suárez et al. [20].

### 2.6. Total Phenol and Total Flavonoid Contents and Antioxidant Activity

Total phenolic compounds (TP) were extracted with an 80% (*v*/*v*) methanol solution. Thus, 1 g of avocado homogenate was mixed with 10 mL of 80% methanol, sonicated in an ultrasonic bath (Conterm 36 L, J.P. Selecta s.a.u., Barcelona, Spain), and centrifuged (Ortoalresa, Microcen 24, Madrid, Spain) at 5000 rpm for 5 min. The filtrate was separated and made up to 10 mL with the extractant solution. Antioxidant activity and total flavonoids were also determined from the methanolic extract.

#### 2.6.1. Total Phenol Content (TP)

The TP in the extracts was measured by the Folin–Ciocalteu reagent (Sigma-Aldrich Chemical Co., St. Louis, MO, USA) method. An aliquot of 1 mL of diluted extract (1:1) was mixed with 1 mL of 1:1 diluted Folin–Ciocalteu phenol reagent (Merck KGaA, Darmstadt, Germany). After 5 min, 2 mL of 10% (*w*/*v*) Na_2_CO_3_ was added, and the mixture was allowed to stand for 10 min and then centrifuged (Ortoalresa, Microcen 24, Madrid, Spain) at 5000 rpm for 5 min. After the colorimetric reaction, the absorbance was measured at 750 nm with a BK-V1000 spectrophotometer (Biobase Biodustry Co., Ltd., Jinan, China). The result was expressed as mg gallic acid (Sigma-Aldrich, St. Louis, MO, USA) equivalent (GAE) 100 g^−1^ fresh sample (FW) (mg GAE 100 g^−1^).

#### 2.6.2. Total Flavonoid Content

The total flavonoid (TF) content was determined using the colorimetric method proposed by Pękal and Pyrzyńska [21]. One milliliter of the methanolic extract was mixed with 75 μL of 5% NaNO_2_ for 5 min. A total of 75 μL of 10% AlCl_3_ solution was added. After 1 min, 500 μL of 1 M NaOH was added, and then the mixture was centrifuged (Ortoalresa, Microcen 24, Madrid, Spain) at 5000 rpm for 5 min. The absorbance was measured at 510 nm with a BK-V1000 spectrophotometer (Biobase Biodustry Co., Ltd., Jinan, China). The result was expressed as mg quercetin (Sigma-Aldrich, St. Louis, MO, USA) equivalent (QE) per 100 g FW (mg QE 100 g^−1^).

#### 2.6.3. Antioxidant Activity

Antioxidant activity was measured using the DPPH (2,2-diphenyl-1-picrylhydrazyl) radical [22] and the ABTS (2,2′-azino-bis(3-ethylbenzothiazoline-6-sulfonic acid) radical [23] methods. DPPH (Sigma-Aldrich, St. Louis, MO, USA) solution (0.1 mmol L^−1^ in methanol) was prepared by dilution with methanol to an absorbance of 1.00 ± 0.01 at 517 nm. An ABTS radical cation solution (ABTS^·+^) was produced by reacting an ABTS (Sigma-Aldrich, St. Louis, MO, USA) stock aqueous solution (7 mmol L^−1^) with 2.45 mmol L^−1^ potassium persulfate followed by incubation in the dark for 16 h at room temperature. The ABTS^·+^ solution was prepared by dilution with ethanol to an absorbance of 0.700 ± 0.005 at 734 nm. Then, 0.3 mL of the above methanolic extract was mixed with 3 mL of DPPH and incubated in the dark for 30 min, after which the absorbance at 517 nm was measured. For ABTS, 0.2 mL of the methanol extract was mixed with 4 mL of ABTS and incubated in the dark for 6 min, after which the absorbance at 734 nm was measured. A blank was prepared in the same manner and measured at 0 min. The absorbance was measured with a BK-V1000 spectrophotometer (Biobase Biodustry Co., Ltd., Jinan, China). The antioxidant capacity was calculated using a calibration curve prepared with Trolox (6-hydroxy-2,5,7,8-tetramethylchroman-2-carboxylic acid) (Sigma-Aldrich, St. Louis, MO, USA) within a range of 200–800 μmol L^−1^, and the result was expressed as µmol Trolox equivalent (TE) per g FW (µmol TE g^−1^).

### 2.7. Fatty Acid Profile and α-Tocopherol Content

To analyze the fatty acid profile (FAME) and α-tocopherol content, fat extraction with n-hexane was performed according to the methods of Meyer and Terry [19] with slight modifications. A total of 1.5 g of dry sample was homogenized with 15 mL of n-hexane employing an Ultra-Turrax T25 homogenizer (Janken and Kunkel Ika-Labortechnik, Staufen, Germany). After 1 min, the mixed was filtered and stored at −20 °C until analysis.

#### 2.7.1. α-Tocopherol Content

An aliquot of this extract in n-hexane was filtered through a 0.45 μm filter, and 0.4 mL was taken and diluted with 1 mL of ethanol/hexane (70:30 *v*/*v*). The sample was then injected into a Waters 2690 HPLC high-performance liquid chromatography (HPLC) instrument equipped with a Waters 2475 fluorescence detector (Waters Corporation, Milford, MA, USA). A Nucleodur 100-5 C18 ec, 4.6 × 250 mm column (Macherey-Nagel, Dueren, Germany) was used for the separation. The mobile phase consisted of methanol (A) and ethanol (B), and an isocratic flow rate of 0.8 mL min^-1^ (80A/20B) was used. The column temperature was set at 30 °C, and the total chromatogram time was 6 min. Peaks were identified by comparing the retention times with those of an α-tocopherol standard (Sigma-Aldrich Chemical Co., St. Louis, MO, USA). The concentration of α-tocopherol was expressed as μg per g FW.

#### 2.7.2. Fatty Acid Profile

For the determination of fatty acid methyl esters (FAME), lipid extracts were transmethylated by acid catalysis [24]. FAMEs were quantified using a TRACE-GC Ultra gas chromatograph (Thermo Fisher Scientific Inc., Waltham, MA, USA) equipped with a column injector; a flame ionization detector; and a fused silica capillary column, Supelcowax TM 10 (30 m × 0.32 mm I.D. × 0.25 μm) (Sigma-Aldrich Co., St. Louis, MO, USA). Helium was used as the carrier gas, and the temperature program was 50–150 °C at a 40 °C min^−1^ slope, then from 150 to 200 °C at 2 °C min^−1^, to 214 °C at 1 °C min^−1^ and, finally to 230 °C at 40 °C min^−1^. Fatty acids were identified and quantified using a reference standard (oil reference standard AOCS, Sigma, St. Louis, MO, USA). Fatty acid composition was expressed as the percentage of total fatty acids.

### 2.8. Statistics

Statistical analysis of the data was performed using SPSS 25.0 (Statistical Package for the Social Sciences) for Windows. The results are expressed as the mean ± standard deviation. Analysis of variance (ANOVA) was applied to all the quantitative variables studied to compare the mean values obtained at the *p* = 0.05 level. The Duncan test was used to classify the values into homogeneous groups. Pearson’s correlation coefficients were calculated to detect relationships between variables.

## 3. Results and Discussion

### 3.1. Physical Characteristics of Hass Avocados

The mean weight and percentage of pulp of Hass avocados grown in Tenerife were 239 ± 54 g and 70.6 ± 4.7, respectively, and these values significantly differed according to location (Figure 2). The mean avocado weights ranged from 185 g to 255 g for avocados harvested in the BU and SA localities, respectively. Avocados from BU also had a greater percentage of pulp (77%) than did those from the RE locality (68%). The percentage of seeds collected from avocados in the BU locality (9%) was lower than that collected in the RE locality (18%). The skin percentage was fairly constant among all localities, with a mean value of 13.6%. These values were similar to those obtained by Jiménez et al. [25], Rodríguez-Carpena et al. [1], Wang et al. [26], Henao-Rojas et al. [27], and Rozan et al. [28]. The differences in fruit weights and percentage of pulp among the locations studied could have been due to a combination of factors, such as soil and edaphoclimatic conditions. The influence of irrigation water stands out since its composition and electrical conductivity affect plants and can cause lower production and lower fruit weight [29]. The locations with the lowest avocado weights (BU and TA) and significant differences from the rest of the locations were those where the irrigation water had a high pH (pH > 9) and a lower availability of Ca since this mineral has little mobility in alkaline soils and the plant suffers from Ca deficiency. Likewise, the locations where the heaviest avocados were detected (OR and SA) were where the pH of the irrigation water was close to 7 (Appendix A).

### 3.2. Proximate Composition

The location of the orchards significantly influenced the contents of fat, fiber, and ash (Table 2). However, Henao-Rojas et al. [27] found no significant differences in fat and ash contents among the eight locations studied but did find significant differences in the crude fiber data. The mean dry matter content (31.7 ± 2.5%) of Hass avocados grown in Tenerife ranged from 29.7% to 33.9%, and the mean oil percentage in avocados was 18.3 ± 2.2%. The dry matter and fat contents of the avocado plants harvested in the SI and SA localities were high, while those harvested in the TA localities had the lowest values for these parameters of all the locations sampled. Dry matter and fat values were within the ranges described for Hass avocados [25,27,30,31,32,33], and fat values were lower than those shown by Daiuto et al. [34], Peraza-Magallanes et al. [35], and Rozan et al. [28]. Avocados were a good source of dietary fiber (6.3 ± 0.7%), with values ranging from 5.4% (BU) to 6.8% (SI). These values were similar to those reported by Henao-Rojas et al. [27] and Hirasawa et al. [36]. The latter authors found that the percentage of soluble fiber was approximately 32%, and that this soluble fiber was composed mainly of soluble pectins and hemicelluloses.

### 3.3. Vitamin C and α-Tocopherol Contents

The Hass avocados analyzed did not have a high vitamin C content, with an average ascorbic acid value of 2.8 mg 100 g^−1^. These findings are consistent with those of Jimenez et al. [22], Luximon-Ramma et al. [37], and Wang et al. [33]. However, Rozan et al. [28] (12.8 mg 100 g^−1^ FW) and Tesfay et al. [7] (13.5–15.6 mg 100 g^−1^ FW) reported much higher ascorbic acid concentrations in Hass avocado pulp. In avocados, α-tocopherol predominates (the most biologically active form of vitamin E), with β-, γ-, and δ-tocopherol being found in lower amounts, and all tocopherols are potent scavengers of lipoperoxyl radicals [11,30,38]. The α-tocopherol content did significantly differ (*p* > 0.05) among the locations studied, in contrast to the findings of Henao-Rojas et al. [27]. Calderón-Vázquez et al. [39] also found no significant differences as a function of the location of production and harvest year, while the intergenotype variances were significant. Avocados did show interesting levels of α-tocopherol, especially those produced in GU, whose mean content (52.2 ± 2.87 µg g^−1^) was more than double that determined at the other locations. These data were similar to those described by Huaman-Alvino et al. [30], Lu et al. [31], Peraza-Magallanes et al. [35], and Cervantes-Paz et al. [40]. In the study by Calderón-Vázquez et al. [39], the range of vitamin E concentrations ranged from 6.98 to 44.9 µg g^−1^ FW for other avocado varieties. Peraza-Magallanes et al. [35] indicated that further research would be necessary to understand whether the differences found in tocopherol contents in the accessions analyzed are due to their racial origin or to environmental conditions. Similarly, Lu et al. [31] reported that the variations in tocopherol content from different locations in the same season were much smaller than the variations between seasons.

### 3.4. Total Phenol and Total Flavonoid Contents and Antioxidant Activity

Hass avocados grown in the GU locality had significantly higher contents (56.0 mg GAE 100 g^−1^ FW) of total phenols (TP) than those grown in the SI (46.0 mg GAE 100 g^−1^ FW) and TA (47.7 mg GAE 100 g^−1^ FW) localities. Our data were within the range found in the literature (expressed in dry weight, DW), which ranged from values below 100 mg GAE 100 g^−1^ DW [1,30,32,37] to values above 300 mg GAE 100 g^−1^ DW [28,33,34]. Huaman-Alvino et al. [30] indicated that the differences in these TP contents could be due to the climatic conditions and agricultural practices in which the Hass avocado are grown. Similarly, Villa-Rodríguez et al. [32] determined that the stage of fruit maturity also influenced the TP content, with the highest TP content occurring when the fruit was ripe. According to Rodríguez-Carpena et al. [1], Rozan et al. [28], and Jimenez et al. [25], among the phenolic compounds, hydroxycinnamic acids predominate, followed by procyanidins, hydroxybenzoic acids, and catechins (epitechatechin). Wang et al. [26] suggested that procyanidins are the phenolic compounds that contribute the most to the antioxidant capacity of avocados.

In the Hass avocados analyzed, total flavonoids (TF) (27.1 ± 6.1 mg QE 100 g^−1^ FW) represented slightly more than 50% of the TP (51.4 ± 8.1 mg GAE 100 g^−1^ FW). This ratio was higher than that shown by Luximon-Ramma et al. [37] and Rozan et al. [28] and lower than that indicated by Villa-Rodríguez et al. [32]. Villa-Rodriguez et al. [32] reported an inverse relationship between the TP and TF contents during the ripening of Hass avocados, such that while the TP increased, the TF decreased slightly. These researchers found that the flavonoid/phenol ratio decreased as avocados ripened.

Regarding the antioxidant capacity of Hass avocados, the results obtained after applying the ABTS method were superior to those determined by the DPPH method, which agrees with those obtained by Daiuto et al. [34] and Rodríguez-Carpena et al. [1]. Previously, several researchers investigated the antioxidant activities of Hass avocados, finding values that ranged remarkably (between 1 and 320 µmol g^−1^ FW for DPPH and between 20 and 940 µmol g^−1^ FW for ABTS). In this work, we found a mean antioxidant capacity for avocados of 1.14 ± 0.36 µmol TE g^−1^ (DPPH) and 3.17 ± 0.38 µmol TE g^−1^ (ABTS), values that were in the range of those described by Luximon-Ramma et al. [37], Wang et al. [26], and Rozan et al. [28]; lower than those reported by Daiuto et al. [34]; and much lower than those shown by Rodríguez-Carpena et al. [1] and Tesfay et al. [7].

### 3.5. Mineral Content

The mineral content of these Hass avocados was important, with values higher than 1.3%, reaching 1.8% in those from the TA locality. These ash percentages were similar to those described in other works [25,28] and higher than those reported by Rodríguez-Carpena et al. [1]. Significant differences (*p* < 0.05) were detected between the different locations for the analyzed minerals, except for Mg, Cu, and Mn. Hass avocados showed high contents of K, followed by P, Mg, and Ca, and presented lower values of Fe, Zn, Cu, and Mn (Table 2). Several locations had somewhat high P contents in fruit, so they had a low Ca content since this mineral will have precipitated due to the soil pH, which in turn is conditioned by the pH of the irrigation water (Appendix A) and could be an indicator for future work. K contents ranged from 4702 mg kg^−1^ FW in avocados produced from the SA locality to 7170 mg kg^−1^ FW in avocados produced from the TA locality. Avocados produced in the TA locality also had the highest P and Zn contents (658 and 6.28 mg kg^−1^ FW, respectively) and the lowest Ca content (66.7 mg kg^−1^ FW). In contrast, those from the RE locality had the highest Ca (125 mg kg^−1^ FW) and Fe (8.73 mg kg^−1^ FW) contents and the lowest P (442 mg kg^−1^ FW) content. The lowest Fe (4.78 mg kg^−1^ FW) and Zn (4.11 mg kg^−1^ FW) contents were detected in avocados from the SA locality. All these data are within the ranges shown in other investigations [27,28,34,41,42,43].

The composition of minerals in fruit pulp is influenced by different factors; for example, Boyd et al. [42] reported that there was much variability in the internal distribution of minerals within the pulp, and that this variability was the same for fruit from both the north and south sides of the tree. Marques et al. [43] reported significant differences in Ca, Mg, K, B, and Zn contents in the pulp of avocados from three different trees and concluded that this variability affected fruit quality among Hass avocado trees. Reddy et al. [41] studied the effect of soil quality on the metal content of fruit pulp and concluded that the plant invariably controlled metal uptake according to its metabolic needs. Additionally, Henao-Rojas et al. [27] indicated that Ca concentration and its relationships with K and N have a significant effect on postharvest fruit softening.

### 3.6. Fatty Acid Profile

The major fatty acids determined in this study (Table 2) were oleic (55.1 ± 3.5%), palmitic (20.1 ± 1.8%), linoleic (12.3 ± 1.7%), and palmitoleic (11.2 ± 1.7%) acids, and linolenic (0.68 ± 0.15%) and stearic (0.54 ± 0.04%) acids were found in lower percentages. In other works, these fatty acids were also found in the majority of Hass avocados [5,6,19,27,30,32,44,45,46,47]. Significant differences in fatty acid profiles were found for the two monounsaturated acids. Thus, avocados from the RE locality showed the highest and lowest percentages of oleic acid (57.4%) and palmitoleic acid (10.1%), respectively, while those from the SA locality showed the lowest percentage of oleic acid (48.7%), and those from the SI locality had the highest percentage of palmitoleic acid (13.9%). In a previous work [5], we found that the fatty acid profile of Hass avocados produced in Tenerife, as well as the total fat and dry matter contents, were influenced by the month, area, and altitude of production. Carvalho et al. [45] reported that the fatty acid content changed with the growing area, and Henao-Rojas et al. [27] also reported that the fatty acid profile was highly related to the geographical region where the orchards were located. Monounsaturated fatty acids (MUFAs) were the main fat component (66.3 ± 2.3%), followed by SFAs (20.7 ± 1.8%) and PUFAs (12.9 ± 1.8%). Henao-Rojas et al. [27] indicated that monounsaturated fatty acids, in addition to being determinants of fruit quality, are related to fruit taste.

### 3.7. Pearson Correlation Analysis

Significant correlations (*p* < 0.05) were obtained between the variables studied. Table 3 shows only the significant correlations between the variables studied. The flesh percentage increased (*p* < 0.001) to a greater extent when the seed percentage decreased (r = −0.849) than when the skin percentage decreased (r = −0.625). The fat percentage increased (*p* < 0.001), as did the dry matter (r = 0.892) and fiber (r = 0.551) contents of Hass avocados. TF correlated positively (*p* < 0.001) with TP (r = 0.624), and both (*p* < 0.001) with the antioxidant capacity (ABTS and DPPH methods) of avocado flesh, with correlation coefficient values between r > 0.59 and r < 0.70. The correlation between the data obtained by the ABTS and DPPH methods was also high (r = 0.678, *p* < 0.001). These correlations have also been found in other works [1,26,33,37] and confirm that phenolic compounds, specifically procyanidins [26], play a fundamental role in the antioxidant action against free radicals. Ash was positively and significantly (*p* < 0.001) correlated with P (r = 0.589) and K (r = 0.708) and negatively correlated with Ca (r = −0.428, *p* =0.002). Among the minerals, P was significantly (*p* < 0.001) positively correlated with K (r = 0.631), Cu (r = 0.530), and Zn (0.668) and negatively correlated with Ca (r = −0.505). Zn was strongly significantly correlated (*p* < 0.005) with K (r = 0.492) and Ca (r = −0.394). P was also correlated with fatty acids, with the exception of palmitic acid. There were significant (*p* < 0.005) and positive correlations of P with palmitoleic (r = 0.406) and linoleic (r = 0.445) acids and negative correlations with oleic acid (r = −0.410). Among the fatty acids determined, those established between oleic acid and palmitoleic acid (r = −0.824) and palmitic acid (r = −0.697) had high correlation coefficients. In addition, there were differences between palmitic and palmitoleic acids (r = 0.596) and between linoleic and oleic acids (r = 0.705).

## 4. Conclusions

The location of the orchards where Hass avocados are grown has a great influence on the chemical composition, so this variation must be considered when establishing requirements for a possible quality mark for Canary Island avocados. Numerous physicochemical parameters, such as weight, percentage of pulp, minerals, fat, fiber, α-tocopherol, flavonoids, and fat profile, significantly differ according to the location of production. The main dry matter component of the Hass avocados studied was fat, followed by fiber. Avocados are a good source of dietary fiber and minerals, especially K, and among the trace elements, Fe. Among the fatty acids, MUFAs, which are the main component of fat, have a high oleic acid content. These findings may contribute to the understanding of the importance of avocados in a balanced and healthy diet. Regarding the content of bioactive compounds, avocados of the Hass variety produced on the island of Tenerife contain high contents of flavonoids, which represent more than 50% of the total phenolic compounds. They also present important contributions of α-tocopherol, especially Hass avocados produced at the GU locality. A large number of significant correlations were found between the parameters studied, such as the correlation between antioxidant capacity and antioxidant compound content (total phenolics and total flavonoids) or the correlation between total dry matter and fat content. The significant influence of orchard location on the chemical composition of Hass avocados highlights the importance of considering environmental and geographical factors in avocado production. This could lead to future research on the relationship between the growing environment and the nutritional quality of agricultural products, which could have important implications for sustainable agriculture and food security.

## Figures and Tables

**Figure 1 foods-13-01058-f001:**
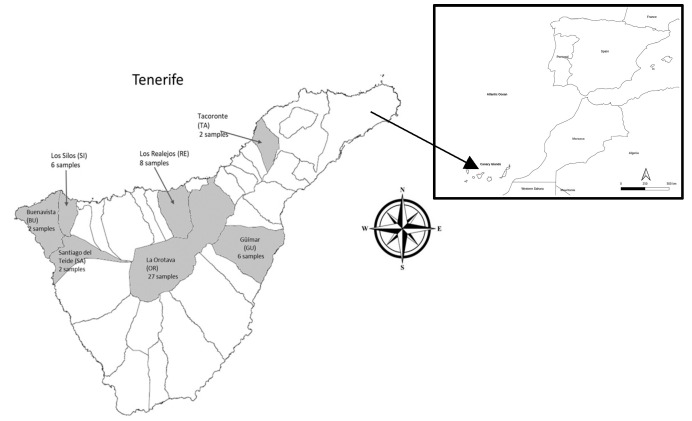
Geographical location of the Canary Islands and map of the location of the Hass avocado orchards studied in Tenerife.

**Figure 2 foods-13-01058-f002:**
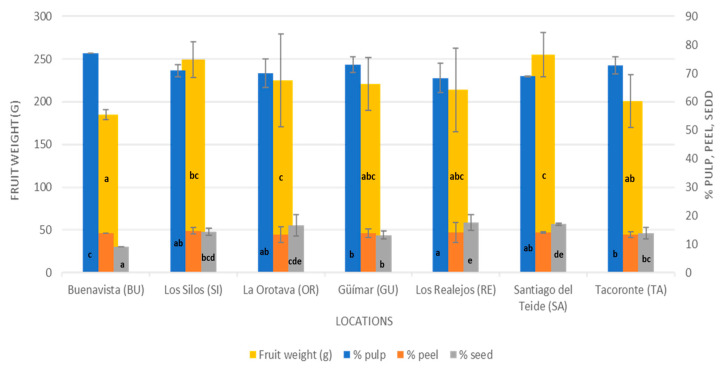
Fruit weight (g) and percentage of pulp, seed, and peel of Hass avocados studied in Tenerife. Different letters in columns indicate significant differences between locations (*p* < 0.05).

**Table 1 foods-13-01058-t001:** Data from the agroclimatic stations.

Agroclimatic Stations	Slope	Length–Latitude	Altitude (msnm)	TM (°C)	Tm (°C)	HRM (%)	HRm (%)	P (mm)	ETo (mm)	Hours Sunshine by Day (h)	N° of Samples	Classification Climatic Papadakis *^,^**
BU	North	28.38136–16.84592	66	26.4	15.1	93.4	43.8	183	1380	9.42	2	Semicálido
SI	North	28.35145–16.80345	450	27.0	12.2	100	30.7	392	969	7.08	2	Tierra templada
OR	North	28.40657–16.51432	214	25.9	13.5	99.9	44.4	2767	973	7.71	27	Tierra templada
GU	South	28.32639–16.40564	280	29.7	13.6	97.9	26.8	102	1407	7.71	6	Tierra templada
RE	North	28.36847–16.57725	595	25.4	9.73	100	31.7	442	821	6.38	8	Tierra templada
SA	South	28.21102–16.82964	30	31.5	12.5	99.6	35.6	58.6	1225	8.87	3	Tierra templada
TA	North	28.46950–16.40043	694	27.6	9.03	96.6	21.3	436	1119	7.68	2	Tierra templada

* Papadakis [15]; ** Santana [16]. TM = average monthly maximum temperature in °C; Tm = minimum monthly temperature in °C; HRM = average monthly maximum relative humidity in %; HRm = average monthly minimum humidity in %; P = total precipitation in the period in mm.; ETo = evapotranspiration in the period in mm. La Orotava (OR), Los Realejos (RE), Tacoronte (TA), Güímar (GU), BU (Buenavista), Los Silos (SI), SA (Santiago del Teide).

**Table 2 foods-13-01058-t002:** Chemical composition of Hass avocado pulp harvested at various locations on the island of Tenerife.

Location	OR	RE	TA	GU	BU	SI	SA
Proximate composition (%)							
Dry matter	31.3 ± 2.25 ^a^	32.1 ± 3.09 ^a^	29.7 ± 2.71 ^a^	32.2 ± 0.97 ^a^	31.2 ± 4.08 ^a^	33.8 ± 1.35 ^a^	33.9 ± 0.23 ^a^
Fat	17.9 ± 1.90 ^ab^	18.3 ± 3.03 ^bc^	16.3 ± 2.46 ^a^	18.7 ± 1.56 b^cd^	19.0 ± 2.08 b^cd^	20.2 ± 0.96 ^cd^	20.4 ± 0.56 ^d^
Fiber	6.32 ± 0.67 ^bc^	6.44 ± 0.37 ^bc^	6.27 ± 0.19 ^bc^	6.03 ± 0.59 ^b^	5.43 ± 1.05 ^a^	6.82 ± 0.50 ^c^	6.39 ± 0.40 ^bc^
Ash	1.45 ± 0.27 ^abc^	1.36 ± 0.25 ^ab^	1.77 ± 0.07 ^d^	1.65 ± 0.21 ^cd^	1.56 ± 0.11 ^bcd^	1.30 ± 0.05 ^a^	1.30 ± 0.12 ^a^
Bioactive compounds and antioxidant capacity							
Ascorbic acid (mg 100 g^−1^)	2.77 ± 1.43 ^a^	2.75 ± 0.59 ^a^	3.26 ± 0.59 ^a^	3.11 ± 1.29 ^a^	2.41 ± 0.14 ^a^	2.81 ± 0.73 ^a^	1.89 ± 0.20 ^a^
α-Tocopherol (µg g^−1^)	20.6 ± 3.59 ^ab^	21.0 ± 4.49 ^ab^	20.8 ± 3.82 ^ab^	52.2 ± 2.87 ^c^	20.7 ± 3.60 ^ab^	18.6 ± 1.29 ^a^	22.8 ± 2.14 ^bc^
Total phenols (mg GAE 100 g^−1^)	52.1 ± 9.96 ^ab^	49.2 ± 5.13 ^ab^	47.7 ± 3.91 ^a^	56.0 ± 5.10 ^b^	48.3 ± 4.09 ^ab^	46.0 ± 2.68 ^a^	53.3 ± 2.68 ^ab^
Total flavonoids (mg QE 100 g^−1^)	27.7 ± 7.42 ^a^	27.5 ± 5.24 ^a^	24.7 ± 3.86 ^a^	24.2 ± 2.74 ^a^	27.8 ± 1.68 ^a^	25.7 ± 3.03 ^a^	30.0 ± 3.31 ^a^
DPPH (µM TE g^−1^)	1.14 ± 0.37 ^a^	1.23 ± 0.39 ^a^	0.92 ± 0.04 ^a^	1.19 ± 0.35 ^a^	1.10 ± 0.29 ^a^	0.87 ± 0.04 ^a^	1.35 ± 0.32 ^a^
ABTS (µM TE g^−1^)	3.21 ± 0.42 ^a^	3.13 ± 0.34 ^a^	3.27 ± 0.18 ^a^	3.26 ± 0.22 ^a^	2.77 ± 0.53 ^a^	3.08 ± 0.11 ^a^	3.12 ± 0.34 ^a^
Minerals (mg kg^−1^)							
Phosphorus	479 ± 10 ^a^	442 ± 72 ^a^	658 ± 84 ^b^	511 ± 80 ^a^	460 ± 58 ^a^	526 ± 104 ^a^	469 ± 67 ^a^
Potassium	5631 ± 896 ^bc^	5261 ± 772 ^ab^	7170 ± 1042 ^e^	6421 ± 415 ^d^	6256 ± 412 ^cd^	5102 ± 799 ^ab^	4702 ± 283 ^a^
Calcium	109 ± 36 ^b^	125 ± 36 ^b^	66.7 ± 20 ^a^	105 ± 19 ^b^	101 ± 22 ^b^	97.4 ± 21 ^b^	101 ± 5 ^b^
Magnesium	352 ± 74 ^a^	350 ± 67 ^a^	312 ± 69 ^a^	352 ± 27 ^a^	298 ± 37 ^a^	340 ± 43 ^a^	350 ± 20 ^a^
Iron	8.00 ± 2.62 ^b^	8.73 ± 1.83 ^b^	8.31 ± 1.48 ^b^	8.25 ± 3.24 ^b^	6.27 ± 1.15 ^ab^	8.50 ± 1.98 ^b^	4.78 ± 0.88 ^a^
Copper	2.74 ± 1.24 ^a^	2.48 ± 0.95 ^a^	2.57 ± 0.48 ^a^	2.80 ± 1.44 ^a^	2.48 ± 0.48 ^a^	3.17 ± 0.48 ^a^	3.99 ± 1.77 ^a^
Zinc	5.74 ± 1.56 ^b^	4.97 ± 0.88 ^ab^	6.28 ± 0.94 ^b^	5.96 ± 1.38 ^b^	5.13 ± 1.61 ^ab^	5.85 ± 1.17 ^b^	4.11 ± 0.26 ^a^
Manganese	1.27 ± 0.37 ^a^	1.42 ± 0.25 ^a^	1.32 ± 0.24 ^a^	1.16 ± 0.13	1.24 ± 0.12 ^a^	1.24 ± 0.17 ^a^	1.20 ± 0.25 ^a^
Fatty acids (% of total fatty acids)							
C16:0 (Palmitic)	20.2 ± 1.67 ^a^	19.6 ± 1.81 ^a^	20.0 ± 2.92 ^a^	19.8 ± 1.92 ^a^	19.5 ± 0.60 ^a^	20.8 ± 1.60 ^a^	23.0 ± 0.47 ^a^
C18:0 (Stearic)	0.53 ± 0.03 ^a^	0.56 ± 0.05 ^a^	0.52 ± 0.04 ^a^	0.57 ± 0.04 ^a^	0.57 ± 0.01 ^a^	0.52 ± 0.03 ^a^	0.55 ± 0.01 ^a^
C16:1 (Palmitoleic)	11.1 ± 1.50 ^ab^	10.1 ± 1.71 ^a^	12.6 ± 1.15 ^abc^	10.9 ± 1.06 ^a^	10.6 ± 0.21 ^a^	13.9 ± 1.27 ^c^	13.4 ± 0.10 ^bc^
C18:1 (Oleic)	55.5 ± 3.47 ^bc^	57.4 ± 2.78 ^c^	53.1 ± 2.33 ^ab^	54.1 ± 2.31 ^bc^	56.9 ± 1.84 ^bc^	51.8 ± 1.49 ^ab^	48.7 ± 3.04 ^a^
C18:2 (Linoleic)	11.9 ± 1.60 ^a^	11.7 ± 1.94 ^a^	13.1 ± 1.49 ^a^	13.9 ± 1.37 ^a^	11.7 ± 2.21 ^a^	12.5 ± 0.62 ^a^	13.8 ± 2.23 ^a^
C18:3 (Linolenic)	0.68 ± 0.16 ^a^	0.68 ± 0.15 ^a^	0.80 ± 0.20 ^a^	0.72 ± 0.09 ^a^	0.60 ± 0.13 ^a^	0.56 ± 0.07 ^a^	0.60 ± 0.20 ^a^
SFAs	20.7 ± 1.69 ^a^	20.2 ± 1.84 ^a^	20.5 ± 2.96 ^a^	20.3 ± 1.94 ^a^	20.1 ± 0.59 ^a^	21.3 ± 1.64 ^a^	23.5 ± 0.48 ^a^
MUFAs	66.7 ± 2.40 ^a^	67.5 ± 1.88 ^a^	65.6 ± 1.22 ^a^	65.0 ± 1.50 ^a^	67.5 ± 1.63 ^a^	65.7 ± 0.97 ^a^	62.1 ± 2.95 ^a^
PUFAs	12.6 ± 1.72 ^a^	12.4 ± 2.08 ^a^	13.9 ± 1.69 ^a^	14.6 ± 1.44 ^a^	12.3 ± 2.33 ^a^	13.1 ± 0.68 ^a^	14.4 ± 2.43 ^a^

The results are expressed as fresh weight. Different letters in a row indicate significant differences between locations (*p* < 0.05). SFAs (total saturated fatty acids), MUFAs (total monounsaturated fatty acids), and PUFAs (total polyunsaturated fatty acids). La Orotava (OR), Los Realejos (RE), Tacoronte (TA), Güímar (GU), BU (Buenavista), Los Silos (SI), Santiago del Teide (SA).

**Table 3 foods-13-01058-t003:** Pearson correlation coefficients (r) of linear regression between the variables studied.

	%peel	%seed	DM	Fat	Fiber	Ash	AA	α-T	TP	TF	DPPH	ABTS	P	K	Ca	Fe	Cu	Zn	Mn	C16:0	C18:0	C16:1	C18:1	C18:2	C18:3	SFA	MUFA	PUFA
Weight	−0.497 **												0.325 *		−0.337 *		0.455 **	0.398 **				0.287 *	−0.324 *				−0.282 *	
%pulp	−0.625 **	−0.849 **				0.284 *											0.283 *											
%peel				0.316 *		−0.285 *		0.357 *		0.340 *					0.338 *										−0.298 *			
%seed					0.281 *																							
DM				0.892 **	0.485 **			0.405 **			0.292 *				0.297 *				0.323 *						−0.299 *			
Fat								0.497 **		0.295 *	0.392 **				0.322 *					0.372 **					−0.472 **	0.374 **		−0.290 *
Fiber																			0.381 **									
Ash													0.589 **	0.708 **	−0.428 **			0.401 **			−0.349 *			0.419 **	0.311 *			0.419 **
AA																				−0.344 *					0.339 *	−0.342 *		
α-T														0.303 *						0.366 **	0.311 *		−0.356 *			0.369 **	−0.342 *	
TP											0.588 **	0.604 **													−0.300 *			
TF											0.696 **	0.600 **	−0.359 *	−0.350 *	0.284 *									−0.333 *	−0.406 **			−0.346 *
DPPH												0.678 **		−0.313 *											−0.398 **			
ABTS																												
P														0.631 **	−0.505 **	0.356 *	0.530 **	0.668 **			−0.342 *	0.406 **	−0.410 **	0.445 **	0.325 *		−0.327 *	0.445 **
K															−0.308 *	0.302 *		0.492 **					−0.313 *	0.346 *	0.360 *		−0.298 *	0.355 *
Ca																		−0.394 **	0.340 *		0.362 **			−0.374 **	−0.336 *			−0.379 **
Fe																	0.379 **	0.392 **	0.327 *									
Cu																		0.370 **		0.331 *		0.295 *	−0.361 **			0.328 *	−0.331 *	
Zn																								0.451 **	0.393 **			0.456 **
Mn																												
C16:0																					0.425 **	0.596 **	−0.697 **		−0.335 *	1.000 **	−0.622 **	
C18:0																										0.442 **		
C16:1																							−0.824 **			0.587 **	−0.525 **	
C18:1																								−0.518 **		−0.693 **	0.915 **	−0.498 **
C18:2																									0.705 **		−0.676 **	0.998 **
C18:3																											−0.352 *	0.745 **
SFA																											−0.622 **	
MUFA																												−0.664 **

Only significant correlations (*p* < 0.05) are shown. ** Correlation is significant at the 0.01 level (two-tailed). * Correlation is significant at the 0.05 level (two-tailed). The variable Mg is not shown because it did not present significant correlations with any other variable. α-T (α-tocopherol), DM (dry matter), AA (ascorbic acid), TP (total phenolics), TF (total flavonoids), SFA (saturated fatty acids), MUFA (monounsaturated fatty acids), PUFA (polyunsaturated fatty acids).

## Data Availability

The original contributions presented in the study are included in the article or Appendix A, and further inquiries can be directed to the corresponding author.

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
