# Peer review of "The Quality Evaluation of Avocado Fruits (Persea americana Mill.) of Hass Produced in Different Localities on the Island of Tenerife, Spain"

_foods, 2024, doi:10.3390/foods13071058_

Round 1

Reviewer 1 Report

Comments and Suggestions for Authors

1. The logic in the abstract section is confusing and needs to be rewritten. The basic framework of the abstract includes research background, existing research problems, research methods, research results, and research significance.

2. Line 23-24: “The avocado belongs to the lauraceae family. Its fruit is botanically a large berry 23 with a large seed inside.” Suggest merging them into one sentence.

3. line 25-26: A significant characteristic of the avocado is that it does not ripen on the tree but 25 ripens after harvest. There is a grammar error in the expression of this sentence, please correct it.

4. Introduction section. What is the current status and progress of research related to the theme of this project, such as whether the Quality Evaluation of Avocado Fruits can be limited to the Island of Tenerife, and how can Avocado Fruits research in other regions be appropriately expanded? Additionally, what are the current limitations of research? I suggested the author to add the above information to the Introduction section.

5. The title of section 2.1 needs to be modified. Suggest using “Plant material”. Delete "Sampling and climatic conditions of the different locations".

6. The number in Table 1, such as "26,4" in the TM (º C) column, it is referring to "264" or "26.4"? Similar issues also exist in other places, please check and modify them.

7. What do the letters a, b, c, etc. in Table 2 represent? Please annotate them in the table.

8. The discussion section in the article has been added to section 3 "result". Therefore, it is recommended to change the title of section 3 to "Results and Discussion". In addition, discussions related to the topic of this study should be further strengthened, such as “Total phenol and total flavonoid content and antagonistic activity” section.

Comments on the Quality of English Language

Quality of English Language need further improvment.

Author Response

Reviewer 1

The logic in the abstract section is confusing and needs to be rewritten. The basic framework of the abstract includes research background, existing research problems, research methods, research results, and research significance.

The abstract has been rewritten.

Line 23-24: “The avocado belongs to the lauraceae family. Its fruit is botanically a large berry 23 with a large seed inside.” Suggest merging them into one sentence.

We have merged these two phrases into one.

line 25-26: A significant characteristic of the avocado is that it does not ripen on the tree but ripens after harvest. There is a grammar error in the expression of this sentence, please correct it.

This error has been corrected.

Introduction section. What is the current status and progress of research related to the theme of this project, such as whether the Quality Evaluation of Avocado Fruits can be limited to the Island of Tenerife, and how can Avocado Fruits research in other regions be appropriately expanded? Additionally, what are the current limitations of research? I suggested the author to add the above information to the Introduction section.

The evaluation of the quality of Tenerife avocados can be extrapolated to other islands producing this fruit in the Canary archipelago, such as La Palma, La Gomera, Gran Canaria and El Hierro. Likewise, this study can complement those carried out in other subtropical locations with young volcanic soils and carbonate alkaline. These sentences have been included in the introduction.

The title of section 2.1 needs to be modified. Suggest using “Plant material”. Delete "Sampling and climatic conditions of the different locations".

The title of this section has been modified according to your suggestions.

The number in Table 1, such as "26,4" in the TM (º C) column, it is referring to "264" or "26.4"? Similar issues also exist in other places, please check and modify them.

Yes, it is a mistake, the correct data is 26.4. The rest of the data has been corrected.

What do the letters a, b, c, etc. in Table 2 represent? Please annotate them in the table.

The different letters in a row indicate significant differences between locations (p < 0.05). It has been corrected in text.

The discussion section in the article has been added to section 3 "result". Therefore, it is recommended to change the title of section 3 to "Results and Discussion".

The error has been corrected and the title of this section is Results and Discussion

In addition, discussions related to the topic of this study should be further strengthened, such as “Total phenol and total flavonoid content and antagonistic activity” section.

The discussion of the results has been improved.

Reviewer 2 Report

Comments and Suggestions for Authors

In my opinion, the manuscript entitled The Quality Evaluation of Avocado Fruits (Persea americana 2 mill.) of Hass Produced on the Island of Tenerife, Spain by Méndez Hernández et al., aimed to characterize and analyze the physicochemical characteristics of different avocado varieties.

I have some comments and suggestions, as follows:

1.        Line 11, please explain under brackets the meaning of TP abbreviation. It would be easier for the readers to understand the abbreviation of total phenols. When an abbreviation it is used for the first time, it is recommended to mention the meaning in brackets. Same for minerals P, K, Ca, Fe, Zn.

2.        Line 23.  lauraceae family must be written in italic, as follows: Lauraceae. Please correct.

3.        Lines 41-45- please mentioned the references from where authors took the information.

4.        Lines 58-71, please add references for the wrote information. 

5.        Line 113. Please mentioned under brackets the meaning of RH abbreviation.

6.        Line 113. How decided authors that the avocado was complete ripened after the storage at 20°C and relative humidity of 95%?

7.        Line 160, total phenol content please mentioned the concentration of Na2CO3 used in the analyses.

8.        Line 209. Please better describe the method of fatty acid profile. For instance, which was the initial and the final temperature of the column for how long was kept the same temperature? Did you need a carrier gas?

9.        At table to, please divide the analyses to be better highlighted. For instance, fruit weight, %pulp, % peel and % seed could be considered physic analysis, meantime, dry matter, fat, fiber, ash could be proximate composition and macro and micro minerals could be also divided. Fatty acids could also have a row in the table with fatty acids, before putting the values. Also, under the table, mentioned the abbreviations used for fatty acids and all compounds.

10.  Line 242. Probably different letters indicate that there are significant differences? 

11.  Please justified the differences identified at mineral avocado profile. Authors compared the obtained values but did not justify the differences between avocado varieties. Probably pedo-climatic condition, chemical soil composition? Same issue at fatty acid profile, total phenols, total flavonoids, vitamin C.

Comments on the Quality of English Language

 Minor editing of English language required

Author Response

Reviewer 2:

1. Line 11, please explain under brackets the meaning of TP abbreviation. It would be easier for the readers to understand the abbreviation of total phenols. When an abbreviation it is used for the first time, it is recommended to mention the meaning in brackets. Same for minerals P, K, Ca, Fe, Zn.

The meaning of abbreviations in parentheses has been included.

2. Line 23.  lauraceae family must be written in italic, as follows: Lauraceae. Please correct.

The name has been correctly spelled in italic.

3. Lines 41-45- please mentioned the references from where authors took the information.

The references have been included.

4. Lines 58-71, please add references for the wrote information. 

Two new references have been included at the end of this text:

Galán, V. Los frutales tropicales en los subtrópicos. I. Aguacate – Mango – Litchi y Longan; Mundi Prensa Libros S.A.: Madrid, 1990.

Rodríguez, L., Cáceres, J.J. Rentabilidad del cultivo del Aguacate en Canarias. Servicio de Agricultura y Desarrollo rural. Área de Aguas, agricultura, Ganadería y Pesca del Excmo. Cabildo Insular de Tenerife; Tenerife, 2014.

5. Line 113. Please mentioned under brackets the meaning of RH abbreviation.

The meaning of the abbreviation in brackets has been included.

6. Line 113. How decided authors that the avocado was complete ripened after the storage at 20°C and relative humidity of 95%?

Avocados were considered ripe when skin color reached color 6, as indicated by White et al. [17].

7. Line 160, total phenol content please mentioned the concentration of Na2CO3 used in the analyses.

The description of the method has been improved and the Na2CO3 concentration has been included.

8. Line 209. Please better describe the method of fatty acid profile. For instance, which was the initial and the final temperature of the column for how long was kept the same temperature? Did you need a carrier gas?

The description of the method has been improved.

9. At table to, please divide the analyses to be better highlighted. For instance, fruit weight, %pulp, % peel and % seed could be considered physic analysis, meantime, dry matter, fat, fiber, ash could be proximate composition and macro and micro minerals could be also divided. Fatty acids could also have a row in the table with fatty acids, before putting the values. Also, under the table, mentioned the abbreviations used for fatty acids and all compounds.

For better understanding, the results have been divided into different sections. The meaning of the abbreviations for all compounds has also been added.

The physical parameters are shown in a figure, so they have been removed from this table.

10. Line 242. Probably different letters indicate that there are significant differences? 

Yes, different letters in a row indicate significant differences between locations (p < 0.05). It has been corrected in text.

11. Please justified the differences identified at mineral avocado profile. Authors compared the obtained values but did not justify the differences between avocado varieties. Probably pedo-climatic condition, chemical soil composition? Same issue at fatty acid profile, total phenols, total flavonoids, vitamin C.

The discussion of the results has been improved.

Reviewer 3 Report

Comments and Suggestions for Authors

The paper deals with different aspects of chemical composition of avocado fruits produced in different areas of the island of Tenerife, belonging to the autonomous community of the Canary Islands.

The paper is interesting but has several opportunities for improvement.

The wording of the abstract can be improved to give a clearer idea about the work.

In l35 it talks about sugar, it should be sugars or maybe carbohydrates.

Paragraph l40-42 should have a reference at the end of the paragraph.

In the introduction when going from the general context to the specific situation of Tenerife, it should be better written and also consider that the reader does not necessarily have knowledge of such specific geographic issues specific to Spain.

Regarding the agro-climatic information it would be interesting to add information about the hours of sunshine.

In the preparation of the samples, specifically the microwave drying method is firstly not adequately described and secondly it is necessary to justify the effects of this drying on the chemical and quality properties. Perhaps it would have been ideal to freeze-dry rather than microwave dry.

It is also not specified to what humidity the samples were dried.

In materials and methods it is also necessary to specify the models of the equipment used.

It is also necessary to properly identify the equipment and models used.

The wording of the different analyses should be improved and standardized, there are methods that are only mentioned and some that are described in detail.

It is important to specify clearly throughout the document that hass avocado was used.

In relation to the raw materials, is there any variable of quality, caliber, color that allows standardizing the experiment.

When describing/identifying the geographical zones, could it be interesting to assign them a number or codification for an easier to understand presentation and discussion of the results?

The discussion in general is very simple and can be improved by addressing more deeply the meaning of the results in their context rather than just presenting the values.

It would be interesting for the authors to think about providing some graphs with results to improve the presentation of the paper. Currently the manuscript has only two tables as results, and Table 3 is very large and somewhat difficult to read.

Concerning the conclusions, it is necessary to give more importance to the contribution of the results in the discipline and also their possible applications and future research.

Author Response

Reviewer 3:

The wording of the abstract can be improved to give a clearer idea about the work.

The abstract has been rewritten.

In l35 it talks about sugar, it should be sugars or maybe carbohydrates.

Ok, changed sugar to carbohydrates.

Paragraph l40-42 should have a reference at the end of the paragraph.

The references have been included.

In the introduction when going from the general context to the specific situation of Tenerife, it should be better written and also consider that the reader does not necessarily have knowledge of such specific geographic issues specific to Spain.

A map showing the geographical location of the Canary Islands is included in Figure 1.

Regarding the agro-climatic information it would be interesting to add information about the hours of sunshine.

The sunshine hours have been listed in Table 1.

In the preparation of the samples, specifically the microwave drying method is firstly not adequately described and secondly it is necessary to justify the effects of this drying on the chemical and quality properties. Perhaps it would have been ideal to freeze-dry rather than microwave dry.

We are of the same opinion, but when we received the samples we did not have the freeze dryer. The analyses of the most temperature-sensitive components were performed on the fresh homogenized sample. Moisture, ash, dietary fiber, total fat and minerals were analyzed in the dry sample.

It is also not specified to what humidity the samples were dried.

The final moisture content of all samples was 4-5%. This has been included in the text.

In materials and methods it is also necessary to specify the models of the equipment used. It is also necessary to properly identify the equipment and models used.

This information has been included in the text.

The wording of the different analyses should be improved and standardized, there are methods that are only mentioned and some that are described in detail.

The total phenol content, total flavonoid content, and fatty acid profile methods have been modified to include more details of the same.

It is important to specify clearly throughout the document that hass avocado was used.

We specify in the title and in the text that the avocado variety studied is Hass.

In relation to the raw materials, is there any variable of quality, caliber, color that allows standardizing the experiment.

Avocados were considered ripe when skin color reached color 6, as indicated by White et al. [17].

When describing/identifying the geographical zones, could it be interesting to assign them a number or codification for an easier to understand presentation and discussion of the results?

The names of the locations have been changed by codes in all the text.

The discussion in general is very simple and can be improved by addressing more deeply the meaning of the results in their context rather than just presenting the values.

The discussion of the results has been improved.

It would be interesting for the authors to think about providing some graphs with results to improve the presentation of the paper. Currently the manuscript has only two tables as results, and Table 3 is very large and somewhat difficult to read.

It is true that it is a bit big table. We have removed the physical parameters from this table and they are shown as a Figure 2.

Concerning the conclusions, it is necessary to give more importance to the contribution of the results in the discipline and also their possible applications and future research.

The conclusions has been improved.

Reviewer 4 Report

Comments and Suggestions for Authors

As the title of the manuscript suggests, it concerns the assessment of the quality of avocados grown in Tenerife, Spain. The obtained values of individual indicators were compared with other studies available as literature sources, but no attention was paid to the origin (place of cultivation) of the fruits with which they were compared. The abstract, conclusions and introduction show that the impact of 7 growing regions in Tenerife on the physicochemical indicators of the tested avocado fruits was also assessed. Hence the manuscript is a bit inconsistent. Moreover, it requires overall improvement, especially in terms of strengthening the discussion of the results obtained. A comparison in such a way that in other publications the values of a given indicator were higher or lower or were within a similar range is not enough for the internationally valued "FOODS" journal.

There are doubts about the selection of a different number of samples from individual growing regions. Only 1 fruit coded "Buenavista" is from another "Classification climatic Papadakis", i.e. Semicálido and the rest, ranging from 3 to 27 fruits, are from Tierra Templada. However, it was shown that this "Buenavista" fruit differed significantly in the values of many analyzed indicators from other fruits. How can the authors be sure of the reliability of the results obtained if only one fruit was tested?

It should be emphasized that the manuscript covers a wide range of research on physicochemical determinations. The authors have prepared the entire manuscript quite concisely. They can be considered specialists in the subject matter, and the research results, after refinement, can be valuable and used scientifically and practically.

The methodology should be supplemented with more information, including how sample extracts were prepared for chemical determinations.

Were all the fruits the same ripeness? How was fruit ripeness assessed?

Lines 121-123 and 130-133:  "Half of the rest of each fruit (without skin and seed) was also homogenized with a mortar and pestle, transferred to a laboratory glass dish and then placed in a microwave oven for drying." and “Moisture was determined by gravimetry by first drying the sample in a microwave oven and then introducing it into an oven (J.P. Selecta, S.a.u., Barcelona, Spain) at 70 ℃ until a constant weight was obtained, and dry matter content was calculated by difference.” - What were the other parameters of microwave drying, such as microwave power, constant microwave operation or intermittently? Was the temperature of 70C measured in the material or at the outlet from the chamber?

Line 156: "Total phenolic compounds (TP) were extracted with an 80% (v/v) methanol solution." - This sentence requires correction. Furthermore, the preparation of the extracts should be described in more detail.

Line 197: “An aliquot of this extract was filtered in n-hexane through a 0.45 μm filter, and 0.4 mL was taken and diluted with 1 mL of ethanol:hexane (70:30 v/v)." - This sentence requires correction.

Lines 220-225: "Analysis of variance (ANOVA) was applied to all the quantitative variables studied to compare the mean values obtained at the p < 0.05 level." - The value of 0.05 is a threshold and values smaller than 0.05 (statistically significant differences) or greater than 0.05 (no statistically significant differences) are considered. Hence, "at the p < 0.05 level." should be corrected. to "at the p = 0.05 level."

The discussion of the results is incomplete. The focus was only on demonstrating the ingredients as the beneficial features of these fruits. Mean values with standard deviation or the range of obtained values were given alternately. Differences in the values of individual indicators within the origin of fruit were not analyzed. A range of values was provided, e.g. total fiber, but the results of statistical analysis were not used in the description of these values.

"In avocado fiber, insoluble fiber predominates over soluble fiber [32]." (line 257) - This is not enough information to interest the reader. A bit more detail is needed. Moreover, the scope of the research included in this manuscript does not include the determination of the fiber fraction, only its total content.

The conclusions and abstract state the existence of a correlation between antioxidant activity and antioxidant compound content, but the manuscript did not determine or analyze antioxidant compound content.

The selection of references is correct, it consists of 42 items, including about 15% from recent years (2020-2023).

The manuscript requires checking the text for the style of some sentences and other minor errors.

Comments on the Quality of English Language

The manuscript requires minor linguistic corrections.

Author Response

There are doubts about the selection of a different number of samples from individual growing regions. Only 1 fruit coded "Buenavista" is from another "Classification climatic Papadakis", i.e. Semicálido and the rest, ranging from 3 to 27 fruits, are from Tierra Templada. However, it was shown that this "Buenavista" fruit differed significantly in the values of many analyzed indicators from other fruits. How can the authors be sure of the reliability of the results obtained if only one fruit was tested?

We have detected an error in the number of samples in Table 1. Thus, in Buenavista and Los Silos, avocados from two orchards were analyzed. I hope you will excuse this error.

It should be emphasized that the manuscript covers a wide range of research on physicochemical determinations. The authors have prepared the entire manuscript quite concisely. They can be considered specialists in the subject matter, and the research results, after refinement, can be valuable and used scientifically and practically.

The methodology should be supplemented with more information, including how sample extracts were prepared for chemical determinations.

The description of the methods of analysis has been improved, including sample preparation.

Were all the fruits the same ripeness? How was fruit ripeness assessed?

Avocados were considered ripe when skin color reached color 6, as indicated by White et al. [17].

Lines 121-123 and 130-133:  "Half of the rest of each fruit (without skin and seed) was also homogenized with a mortar and pestle, transferred to a laboratory glass dish and then placed in a microwave oven for drying." and “Moisture was determined by gravimetry by first drying the sample in a microwave oven and then introducing it into an oven (J.P. Selecta, S.a.u., Barcelona, Spain) at 70 ℃ until a constant weight was obtained, and dry matter content was calculated by difference.” - What were the other parameters of microwave drying, such as microwave power, constant microwave operation or intermittently? Was the temperature of 70C measured in the material or at the outlet from the chamber?

Analyses of ascorbic acid, total phenolics, total flavonoids, antioxidant activity, moisture, α-tocopherol and fatty acid profile were performed on fresh samples, and the rest of the analyses were performed on dry samples.

Moisture determination was carried out by the air oven drying method at 70°C until constant weight. First in a microwave oven, to accelerate the elimination of water, and then the sample was dried in an air oven at 70°C until a constant weight was obtained.

A portion of the homogenized fresh sample was dried, independently of the moisture determination, to obtain a dry sample.

Line 156: "Total phenolic compounds (TP) were extracted with an 80% (v/v) methanol solution." - This sentence requires correction. Furthermore, the preparation of the extracts should be described in more detail.

The preparation of the extracts in more detail has been included in the text.

Line 197: “An aliquot of this extract was filtered in n-hexane through a 0.45 μm filter, and 0.4 mL was taken and diluted with 1 mL of ethanol:hexane (70:30 v/v)."  This sentence requires correction.

The sentence has been corrected and more details on the preparation of the extracts have been included in the text.

Lines 220-225: "Analysis of variance (ANOVA) was applied to all the quantitative variables studied to compare the mean values obtained at the p < 0.05 level." - The value of 0.05 is a threshold and values smaller than 0.05 (statistically significant differences) or greater than 0.05 (no statistically significant differences) are considered. Hence, "at the p < 0.05 level." should be corrected. to "at the p = 0.05 level."

This has been corrected in the text.

The discussion of the results is incomplete. The focus was only on demonstrating the ingredients as the beneficial features of these fruits. Mean values with standard deviation or the range of obtained values were given alternately. Differences in the values of individual indicators within the origin of fruit were not analyzed. A range of values was provided, e.g. total fiber, but the results of statistical analysis were not used in the description of these values.

The discussion of the results has been improved.

"In avocado fiber, insoluble fiber predominates over soluble fiber [32]." (line 257) - This is not enough information to interest the reader. A bit more detail is needed. Moreover, the scope of the research included in this manuscript does not include the determination of the fiber fraction, only its total content.

It is true that in our work we did not analyze the fiber fractions, so we thought it was interesting to include this reference because it is the only one where the soluble and insoluble fiber contents were analyzed. In addition, they also showed the composition of soluble fiber. This has been included in the text.

The conclusions and abstract state the existence of a correlation between antioxidant activity and antioxidant compound content, but the manuscript did not determine or analyze antioxidant compound content.

The antioxidant components determined in this work are total phenols and flavonoids. There is a correlation between these components and the antioxidant capacity of avocado. We have modified the abstract and conclusions to clarify this point.

The selection of references is correct, it consists of 42 items, including about 15% from recent years (2020-2023). 

In order to improve the discussion of the results, we believe it is necessary to include other references, up to a total of 47 items.

The manuscript requires checking the text for the style of some sentences and other minor errors.

Errors have been corrected throughout the text and in the tables.

Round 2

Reviewer 1 Report

Comments and Suggestions for Authors

The author has made revisions to my suggestions.

Author Response

This reviewer did not comment.

Reviewer 2 Report

Comments and Suggestions for Authors

In my opinion, authors have highly improved the entitled article The Quality Evaluation of Avocado Fruits (Persea americana  mill.) of Hass Produced on the Island of Tenerife, Spain and now it could be published in the present form.

Comments on the Quality of English Language

 Minor editing of English language required

Author Response

Some sentences of the paper were improved.

Reviewer 3 Report

Comments and Suggestions for Authors

The authors made most of the proposed changes and suggestions so I agree with the current version of the document.

Author Response

This reviewer did not comment on the new version of the paper. 

Reviewer 4 Report

Comments and Suggestions for Authors

The manuscript is corrected, but I would suggest that the authors consult it as a whole again, look at the title, possibly correct it, precisely define the purpose of the research, and try to correct some parts of the manuscript according to the title and the stated goal. It is worth emphasizing the very wide scope of research carried out in terms of the properties of avocado fruits and information about the microclimate of the regions of their cultivation.

The authors tried to provide a more in-depth discussion of the results obtained.

However, the purpose of research is still not well defined. In the Introduction, lines 60-80, it is ambiguously stated why fruit from different areas of Tenerife were tested, and the added sentences (lines 75-80) are not accurate. This whole part is a bit disjointed.

The abstract mentions the influence of orchard location on differences in fruit properties but omits the connection with the influence of microclimate conditions characterizing individual regions. The conclusions only state that there is an influence of location, but do not specify what it is. However, the most important thing is still the value of the marked ingredients and the correlation of their content, e.g. with antioxidant activity.

These final corrections should not be that big anymore. However, it is impossible to ignore the microclimate conditions of individual regions, since so much information has been collected about it. It may be worth considering adding principal component analysis PCA to clarify these issues.

I still have great doubts about whether even 2 fruits are enough to assess their quality in the entire orchard.

Lines 86-90: I suggest completing the coding of the location of avocado fruits in the same way as in the discussion of the results, i.e. by adding "locality".

Lines 118-119: “The irrigation water for each location is shown in Supplementary Table 1.” – In this sentence, it is worth adding what was irrigated, i.e. the Hass avocado orchards.

The conclusions should be further refined because there is no information on which avocado fruits (orchard locations) differ the most from each other and what these differences are.

Comments on the Quality of English Language

The quality of the English language is quite good.

Author Response

The manuscript is corrected, but I would suggest that the authors consult it as a whole again, look at the title, possibly correct it, precisely define the purpose of the research, and try to correct some parts of the manuscript according to the title and the stated goal. It is worth emphasizing the very wide scope of research carried out in terms of the properties of avocado fruits and information about the microclimate of the regions of their cultivation.

We have changed the title according to your suggestions

The authors tried to provide a more in-depth discussion of the results obtained.

However, the purpose of research is still not well defined. In the Introduction, lines 60-80, it is ambiguously stated why fruit from different areas of Tenerife were tested, and the added sentences (lines 75-80) are not accurate. This whole part is a bit disjointed.

In the Introduction, we have added two paragraphs to clarify the objective of our study:

As a plant that requires tropical or subtropical climatic conditions, avocado grows well in the Canary Islands, with the Hass cultivar being the main one in the local market. The rate of avocado production in Tenerife is increasing; between 2007 and 2021, the cultivation area tripled, reaching 956 ha and representing 42% of the total cultivation area of this plant in the Canary Islands [12]. To date, the Canary Islands avocado is mainly present in the local market, but it is highly appreciated abroad for its flavor. The purpose of this study is a first step to establish the composition of avocados produced in Tenerife, to promote the consumption and export of the Canary Hass avocado in Europe and to create a quality brand to differentiate the Canary Island Hass avocado from those produced in competing countries. The island of Tenerife has numerous microclimates where avocados are grown, both on the northern and southern slopes and from sea level to almost 1000 m altitude. This means that avocados produced in different parts of the island have different characteristics and qualities depending on the growing areas. There are no clearly established criteria for the quality of Hass avocado produced in Tenerife. Therefore, in this study, the physicochemical characteristics and composition of Hass avocados from orchards located in different agroclimatic conditions of Tenerife were determined. The characteristics of the island of Tenerife, which has different cultivation heights and slopes, resemble those of a small continent, and its location in the subtropics means that the results can be extrapolated to the production of avocados at other latitudes in the face of climate change [13,14]. The evaluation of the quality of Tenerife avocados can be extrapolated to other islands producing this fruit in the Canary Island, such as La Palma, La Gomera, Gran Canaria and El Hierro. Likewise, this study can complement those carried out in other subtropical locations with young volcanic soils and carbonate alkaline waters from the recharge of mountain aquifers.

The abstract mentions the influence of orchard location on differences in fruit properties but omits the connection with the influence of microclimate conditions characterizing individual regions. The conclusions only state that there is an influence of location, but do not specify what it is. However, the most important thing is still the value of the marked ingredients and the correlation of their content, e.g. with antioxidant activity.

Thank you very much for your clarifications. In the results, mention has been made of the relationship with the locations, expressly thinking about their interest for the Protected Geographical Indication (PGI) "Avocado from the Canary Islands", currently being processed in the European Union.

These final corrections should not be that big anymore. However, it is impossible to ignore the microclimate conditions of individual regions, since so much information has been collected about it. It may be worth considering adding principal component analysis PCA to clarify these issues. I still have great doubts about whether even 2 fruits are enough to assess their quality in the entire orchard.

We appreciate your recommendations very much. When we performed the statistical treatment of the results we also applied multivariate analysis to the total number of samples (50) and variables analyzed by location, but did not obtain satisfactory results. In the case of PCA 9 factors were extracted, which explained 79.5% of the variance, and the variables associated with each factor had a good explanation (e.g. Factor 1 was associated mainly with K, Ash and P, Factor 2 with phenols, flavonoids and antioxidant capacity; Factor 3 with the major fatty acids in avocado, ....) but then in the graphical representation of these factors differentiating the samples according to the 7 locations we could not draw any conclusion because there was a great dispersion among them. Another analysis we also performed was discriminant analysis. In this case, after applying the stepwise method, only 48.9% of the avocado samples were correctly classified according to the locality where they were collected. For all these reasons, it was decided not to include these analyses in the text.

We agree with the reviewer that two samples is not the ideal number, in fact initially more were planned, but finally only two samples could be taken. However, we consider that they are sufficient to assess this area since they are not two fruits, but two samples each from two orchards, where three different trees were sampled and three fruits were collected, making a total of 9 avocados per orchard.

Lines 86-90: I suggest completing the coding of the location of avocado fruits in the same way as in the discussion of the results, i.e. by adding "locality".

Ok, we have changed locations to localities.

Lines 118-119: “The irrigation water for each location is shown in Supplementary Table 1.” – In this sentence, it is worth adding what was irrigated, i.e. the Hass avocado orchards.

Ok, the sentence has been corrected.

The conclusions should be further refined because there is no information on which avocado fruits (orchard locations) differ the most from each other and what these differences are.

Variations in the composition and physical parameters of avocados according to localities are described in the results and summarized in the abstract; to include them also in the conclusions would increase the length of this section too much.
